# Comparing the Indian Autism Screening Questionnaire (IASQ) and the Indian Scale for Assessment of Autism (ISAA) with the Childhood Autism Rating Scale–Second Edition (CARS2) in Indian settings

Satabdi Chakraborty[1]☯, Triptish Bhatia[ID][2]☯, Nitin Antony[3], Aratrika Roy[3], Vandana Shriharsh[4], Amrita Sahay[5], Jaspreet S. Brar[6], Satish Iyengar[7], Ravinder Singh[8], Vishwajit L. Nimgaonkar[9], Smita Neelkanth Deshpande[10]¤*

1 Department of Psychiatric Social Work, Centre of Excellence in Mental Health, Atal Bihari Vajpayee Institute of Medical Sciences-Dr. Ram Manohar Lohia Hospital, New Delhi, India, 2 Indo-US Projects, Department of Psychiatry and De-addiction, Centre of Excellence in Mental Health, Atal Bihari Vajpayee Institute of Medical Sciences-Dr. Ram Manohar Lohia Hospital, New Delhi, India, 3 Development and Validation of the Screening Version of ISAA, 'ICMR Project', Dept. of Psychiatric Social Work, Centre of Excellence in Mental Health, Atal Bihari Vajpayee Institute of Medical Sciences-Dr Ram Manohar Lohia Hospital, New Delhi, India, 4 Department of Clinical Psychology, Centre of Excellence in Mental Health, Atal Bihari Vajpayee Institute of Medical Sciences-Dr. Ram Manohar Lohia Hospital, New Delhi, India, 5 National Institute for the Empowerment of Persons with Intellectual Disabilities (NIEPID), Noida, U.P., India, 6 Department of Psychiatry and Consultant, Community Care Behavioral Health Organization, Western Psychiatric Hospital of UPMC, Pittsburgh, Pennsylvania, United States of America, 7 Department of Statistics, University of Pittsburgh, Pittsburgh, Pennsylvania, United States of America, 8 Division of Non Communicable Diseases, Indian Council for Medical Research, New Delhi, India, 9 Department of Psychiatry and Department of Human Genetics, University of Pittsburgh School of Medicine and Graduate School of Public Health, Pittsburgh, Pennsylvania, United States of America, 10 Department of Psychiatry, Centre of Excellence in Mental Health, Atal Bihari Vajpayee Institute of Medical Sciences-Dr. Ram Manohar Lohia Hospital, New Delhi, India

☯ These authors contributed equally to this work.
¤ Current address: St John's Research Institute, St John's National Academy of Health Sciences, Bengaluru, India
* smitadeshp@gmail.com

**Data Availability Statement:** Data are all contained within the paper.

## Abstract

The Indian Autism Screening Questionnaire (IASQ), derived from the Indian Scale for Assessment of Autism ISAA (the mandated tool for autism in India), is an autism screening instrument for use in the general population by minimally trained workers. While ISAA has 40 items with four anchor points, the IASQ is a 10-item questionnaire with yes/no answers. It was initially validated using the ISAA. During its development the ISAA was itself compared to the Childhood Autism Rating Scale version 1 (ISAA Manual). In the present study, we evaluated both the ISAA and the IASQ in relation to the Childhood Autism Rating Scale version 2 (CARS-2). **Methods:** Participants were recruited from three settings: a referral clinic for neurodevelopmental conditions run by the Department of Paediatrics of a tertiary care teaching hospital (NDC OPD), the outpatient department of an institute for disability and rehabilitation (NIEPID), and from the community (CGOC). Persons between ages 3–18 were recruited following consent or assent (parent and child/adolescent). The IASQ was

**Funding:** This research was funded by the Indian Council of Medical Research Task Force for Capacity Building for National Mental Health Programme (File No No. 5/4-4/151/M/2017/NCD-I) and the Fogarty International Centre, NIH ('Cross-Fertilized Research Training for New Investigators in India and Egypt', D43 TW009114 Fogarty International Centre, NIH and HMSC File No. Indo-Foreign/35/M/2012-NCD-1).The content of this manuscript is solely the responsibility of the authors and does not necessarily represent the official views of Fogarty International Centre, NIH (Cross-Fertilized Research Training for New Investigators in India and Egypt' (D43 TW009114) or ICMR. The funding agencies had no role in the design and conduct of the study; collection, management, analysis, and interpretation of the data; preparation, review, or approval of the manuscript; and decision to submit the manuscript for publication.

**Competing interests:** The authors have declared that no competing interests exist.

administered by a minimally trained administrator. It was followed by ISAA and the CARS-2 (in alternating order, by different evaluators blind to each other) (CARS2 SV (Standard Version) and CARS2 HF (High Functioning) as applicable). Sensitivity, specificity and area under the Receiver Operator Characteristics (ROC) curve were calculated for IASQ and CARS2, as well as for ISAA and CARS2. Concordance between CARS2 and ISAA was calculated using kappa coefficient. **Results:** A total of 285 participants (NIEPD n = 124; NDC OPD, n = 4; CGOC n = 157) (a total of 70 with autism and 215 controls) participated. IASQ and CARS2 were administered on 285 participants, while IASQ and ISAA were administered on 264 participants. When IASQ was compared to CARS2, sensitivity was 97%, specificity 81%, PPV 63%, NPV 99% at cut off 1 while these values were 97%, 92%, 79% and 99% respectively at cut off 2. There was high concordance between CARS2 and ISAA (Kappa 0.907, p<0.0001). **Conclusions:** IASQ has satisfactory sensitivity, specificity and concordance when compared with CARS2; it can be used for screening children with autism in community. The ISAA also showed a high concordance with CARS2, as it had with the older version of CARS.

## Introduction

Autism imposes a significant public health burden in South Asia. In Bangladesh, Sri Lanka and India the prevalence of autism is about 1:93 children [1] but rigorous population prevalence studies in India are scarce [2]. Among Indian children aged 1–18 years, pooled prevalence of autism in rural areas is 0.11%, while in urban areas it is 0.09% (ages 1–18 years) [2]. Globally, while autism causes more than 58 disability adjusted life-years (DALYs)/100 000 population, other autism spectrum disorders (ASDs) cause 53 DALYs/ 100 000 [3].

Early detection and intervention greatly improve outcomes [4] including employment [5]. However, these presuppose early and easy identification of autism and subsequent expert referral and evaluation [6, 7]. In most countries community identification falls to minimally educated frontline workers. Hence screening instruments need to be short, simple, easy to administer yet reasonably accurate. In India, since awareness of autism is low, and the Indian Rights of Persons with Disability Act offers significant disability benefits, the screening tool should be usable in adults too [8].

The Indian Autism Screening Questionnaire (IASQ) was developed and tested in the Department of Psychiatry of a tertiary care teaching hospital. It was derived from and compared to the Indian Scale for Assessment of Autism (ISAA- the validated tool for detailed assessment of autism in India) [9–11]. The IASQ is a 10-item questionnaire with yes/no answers, specificity and sensitivity at cut off 1 are 99% and 62%, and at cut off 2 are 98% and 71%. The ISAA is a 40 item 5-point Likert scale instrument which requires training, even for qualified personnel. The IASQ is shorter, has explanatory indicative questions as well, and requires relatively brief training that can be provided online [12].

The ISAA was originally tested against the Childhood Autism Rating Scale, 15 item version 1 (2009) [13]. In 2010, the CARS was revised and a second version published [14]. CARS-2 has two versions—standard and high functioning—for brief yet comprehensive summary information about children with autism. It is not a screening instrument, but is useful for differentiating between children with ASD and other cognitive deficits; it can be used by clinicians, parents, teachers and researchers. It requires 20 to 30 minutes for administration [15–17].

The present study was undertaken to compare the IASQ and the ISAA with the CARS2. The study was carried out in non-psychiatric settings and in a community.

## Materials and methods

Permission was obtained from the Institutional Ethics Committee of this Institution and the National Institute for the Empowerment of Persons with Intellectual Disabilities (Divyangjan) (NIEPID), NOIDA, UP, India.

### Setting

The participants for the study were recruited from three different settings. The first was the Neurodevelopmental Clinic (NDC), conducted weekly by the Department of Paediatrics at a free government teaching hospital where attendees are either referred or attended on their own initiative.

The second setting was the outpatient department of the National Institute for the Empowerment of Persons with Intellectual Disability (NIEPID) Model Special Education Center, NOIDA, Uttar Pradesh. The NIEPID is an evaluation and rehabilitation facility under the Department of Empowerment of Persons with Disabilities (Divyangjan), Ministry of Social Justice & Empowerment, Government of India. It is a referral centre for all of Delhi and surrounding areas for state-of-the-art rehabilitation interventions [18].

The third setting was a house-to-house survey in a Central Government Officers Colony (CGOC) New Delhi. There are 376 (main) and approx. 376 (subsidiary) housing units in CGOC, comprising of two types of houses. The main housing unit is inhabited by higher income, better educated officers and their families (higher- and highest-level officers in D2, D1, C2 type units) (which we identified as Officer's Housing Unit (OHU)). Each officer's housing unit has at least one smaller attached subsidiary unit designated as Servants' Housing Unit (SHU).

### Sample

Recruitment was carried out between September 2019 to March 2020 and thereafter from October 2020 to February 2021. The intervening pause in data collection was necessary due to the COVID-19 pandemic lockdown and restrictions. The NDC participants were recruited from the once-weekly clinic, research personnel visited the NIEPID twice a week and the CGOC survey was carried out at least three days per week. The OHU and attached SHU were considered one single unit and unit numbers were selected on the basis of an online table of random numbers. If the resident in a randomly selected unit declined to participate, the next one in line was approached. The sample consisted of 70 children with autism, and 215 children without autism (Table 1). The diagnosis was ascertained by CARS2 ST and HF versions. CARS2 diagnosis fulfills DSM V criteria.

**Table 1. Sociodemographic profile of the sample (n = 285).**

| Variable | Category | Autism (n = 70) | No—Autism (n = 215) | $X^2$/t-score | p-value |
|---|---|---|---|---|---|
| Gender | M/F | 53/17 | 122/93 | 8.018, | 0.005 |
| Age | Mean ± SD | 7.84 ± 3.54 | 11.95 ± 3.97 | 7.709 | <0.00001 |
| School years | Mean ± SD | 1.09±2.48 | 5.19±3.97 | 8.134 | <0.00001 |
| Currently studying | Yes/No | 61/9 | 200/15 | 2.37 | 0.139 |
| Type of school | Regular/Special/Integrated | 30/22/9 | 176/7/17 | 60.21 | <0.00001 |
| Annual family income | Mean ± SD | 54566.67 ± 71386.12 | 25784.48 ± 32752.81 | 4.61 | <0.00001 |
| | Median | 35000 | 12500 | | |

The IASQ was administered to parents/ caregivers, whereas the ISAA and CARS2 were completed based on information from the parents/ caregivers and observations of the child participants. Each participant received Rs. 100/- for participation (approximately US $ 1.50).

## Inclusion and exclusion criteria

Parents of individuals between 3–18 years of age who were willing to participate and provided written informed consent (as well as children who could assent) were included. Children with other comorbid psychiatric disorders, for example psychotic disorders, ADHD and intellectual disabilities were excluded. An asessment of illness severity was not available on most participants and therefore not analyzed.

## Tools used

Sociodemographic data sheet, developmental milestones information sheet (both from the ISAA manual), Indian Autism Screening Questionnaire (IASQ) [11], the Indian Scale for the Assessment of Autism (ISAA) [19] and Childhood Autism Rating Scale 2 (CARS2) [14] were administered to all the participants.

1. Sociodemographic data sheet comprised basic identification details like ID numbers, age and sex of the child and parents, years of education of parents and children, parental occupations, mother tongue, language spoken at home, family history of any mental illness, consanguinity and age of onset of ASD.

2. Developmental milestones information sheet elicited pregnancy details (nature of delivery, pre/peri/post-natal complications) and developmental history (ages of attainment of motor and speech milestones), presence of regression from previously attained developmental milestones, and birth order.

3. Indian Autism Screening Questionnaire (IASQ) [11] is a 10-item simple, easy to use screening tool to identify possible cases in the community. It was developed as a part of the ICMR Task Force Capacity Building projects and is designed for individuals aged 3 to 18 years. It is scored after interaction with the caregiver with or without observation of the child/person in question. Its 10 items require a YES or NO response. It takes approximately 10 minutes to administer. If more than one question is answered as 'yes' then the person needs to be referred to a specialized centre for detailed evaluation.

4. Indian Scale for Assessment of Autism (ISAA), developed by the National Institute for Mentally Handicapped (NIMH) in 2009 is an authorised instrument for certifying disability in persons with autism [19, 20]. It is a 40-item tool subdivided into 6 domains namely social relationship and reciprocity, emotional responsiveness, speech, language and communication, behaviour patterns, sensory aspects and cognitive component. It is rated on a 5-point Likert scale ranging from 1 (rarely) to 5 (always). Scoring is done by observation, clinical evaluation of behaviour, testing by interaction with the subject and information supplemented by parents or caretakers. Administration time is approximately 40 minutes.

5. Childhood Autism Rating Scale-2nd Edition (CARS2) [14] is a new version of the Childhood Autism Rating Scale. It is a clinician rated, validated scale designed to provide brief yet objective and quantifiable ratings based on direct behavioural observation of children with autism including high functioning children aged 2 or older. It consists of three rating forms namely, CARS2 Standard Version (CARS2-ST), CARS-2 High-Functioning Version (CARS2-HF) and CARS2 Questionnaire for Parents and Caregivers (CARS2-QPC). There are 15 items, each rated on a scale of 1 (behaviour within normal limits for that age) to 4

(behaviour severely abnormal for that age), with mid-points in between (1.5, 2.5 and 3.5) to be used when the behaviour appears to fall between two categories. CARS2 has two rating forms (Standard–ST, and High Functioning- HF). Children are rated on CARS2-ST if age is less than 6 years or if IQ is 79 and below or with notably impaired communication and on CARS2-HF forms if IQ is 80 or above or with fluent communication or age is above 6 years [21] The CARS2-HF is used to assess individuals with IQ 80 and above or who have relatively good verbal skills or above age 6 years. CARS2 was used after due permission from its authors.

## Procedure

**Neuro-Developmental Clinic of Department of Paediatrics (NDC).** The NDC is conducted once weekly and attended by 25–30 patients. The attending paediatricians (a Paediatrics consultant and a Senior Resident) referred eligible and willing participants to an IASQ trained MPhil student who informed parents about the study in detail, obtained written informed consent/assent, and administered the IASQ. They were then directed to the research personnel in the next room who separately and sequentially administered the demographic and development form, followed by either ISAA and CARS2 in turn, blind to each other and to the participant's IASQ score. Successive participants were administered either the ISAA first, or the CARS2 first. Participants had been assessed prior to their NDC visit on IQ; all had average IQ scores.

**National Institute for the Empowerment of Persons with Intellectual Disability (NIEPID).** A procedure similar to the NDC was followed, except that the IQ test and subsequently IASQ (if IQ was within normal limits) was administered by a psychologist employed at the institution. Those with IQ scores within normal limits were referred to the research personnel after oral consent. After written informed consent, the ISAA and CARS2 were administered by research personnel trained in its administration blind to each other and to the IASQ score. If IQ of the child was 80 or above, or they have good verbal skills or above the age of 6 years then CARS2-HF was administered in place of CARS2-ST as described in CARS2 manual [21].

**Central Government Officers Colony Community (CGOC).** The Central Public Works Department (CPWD) provided a list of 400 CGOC housing Units (each unit consisting of an Officer's Housing Unit- OHU—and one or two attached Servant Housing Unit/s- SHU) (some were vacant or unoccupied). Repeated messages about the survey and appeals for cooperation were made on the colony Resident Welfare Association WhatsApp groups. House numbers were selected from an online table of random numbers. Data was collected from both OHU and SHUs. If a participant refused, the next housing unit in line was approached. The aim was to recruit as many participants as possible (see sample size calculation below).

Data were collected thrice a week for 3 months till lockdown began in India in March 2020 (200 house units) and thereafter again from October 2020 to February 2021. Each house was approached individually. Following self-introductions and briefings of the nature and purpose of the study, willingness to participate and written informed consent was obtained by a volunteer undergraduate medical student or research personnel. Basic sociodemographic and developmental milestones data and the IASQ were administered/obtained by one researcher while the other administered the ISAA or the CARS-2, each remaining blind to the scores on the other scale. Participants were paid Rs 100/- for their participation.

## Training in scale administration

M.Phil. Psychiatric Social Work students, one undergraduate medical student, and a psychologist at NIEPID were trained in the administration of IASQ. The training included a briefing

about autism, mock IASQ administration, viewing a training video of ISAA items (on which the IASQ is based) familiarisation with individual IASQ items and its supplementary/explanatory questions meant to aid the data collection process [22]. The time required was approximately one hour.

Three Research Fellows of the project holding M.Phil. degree in Psychiatric Social Work (2) and Clinical Psychology (1) were trained in ISAA and CARS2 rigorously over a two-week period by a Rehabilitation Council of India registered Clinical Psychology faculty (VS).

## Sample size calculation

The sample size for community sample was determined using pre-prepared tables of sample size calculation for sensitivity by Bujang and Adnan [23]. Prevalence of Autism was taken as 10% based on Chauhan et al. [2], the null sensitivity was taken as 50% and targeted sensitivity was selected as 90%. A sample size of 120 provided 88.9% power. So, in each community group minimum 120 participants needed to be included.

**Data analysis.** Data obtained were entered into a database specially prepared for the purpose by the Indian Council of Medical Research (ICMR) [24, 25]. Data were cleaned and checked for outliers.

**Test psychometrics.** Sensitivity, specificity and area under the ROC curve were calculated using SPSS version 21 [26]. Positive and negative predictive values (PPV, NPV) were calculated by the formulae: PPV = Number of true positive / (Number of true positives + Number of false positives), NPV = Number of true negative / (Number of true negative + Number of false negative), Positive Likelihood Ratio LR+ was calculated by using the formula: Sensitivity/(1-Specificity).

Likelihood ratios provide the best guide to clinicians to determine whether a diagnostic test usefully changes the odds that a disease exists. By convention, a positive diagnostic likelihood ratio (LR+) between 1 to 5 is associated with a small increase in post-test odds whereas values between 5 and 10 with a moderate increase and values above 10 with a large increase in post-test odds. Both ranges, 5 to 10 and >10 are considered acceptable for clinical decision making. To further examine the effects that these LR+s may have on the likelihood of disease a Fagan's Nomogram may be used after first converting the pre-test odds to a pre-test probability and then drawing a straight line from the pre-test probability through the LR+ to the corresponding post-test probability [27, 28]. Alternatively, interactive online calculators, for example the diagnostic test calculator available at the University of Illinois, Chicago, USA (http://araw.mede.uic.edu/cgi-bin/testcalc.pl) may be used to calculate LR+ and post-test probabilities [29]. The latter two steps were not followed in this study.

Concordance was calculated using kappa value, and ROCs were computed with various cut offs of IASQ and CARS2 diagnoses both Standard (ST) and High functioning (HF).

## Results

### Samples

A total of 411 (NIEPID, n = 157; NDC OPD, n = 24; CGOC, n = 230) children were referred as on February 28, 2021. Out of these 285 participants (NIEPD, n = 124; NDC OPD, n = 4; CGOC n = 157) were included in the study after informed consent/assent and meeting inclusion/exclusion criteria. IASQ and CARS2 scores were available for all consenting 285 participants. IASQ and ISAA scores were available for 264 of the 285 participants. There were 70 children with autism and 215 in the no-autism group (demographic details in Table 1). CARS2 ST was administered on 121 participants and CARS2 HF on 164 following the CARS2 manual. However, for sensitivity and specificity analyses, ST and HF participants were combined as diagnosis can be made from either form of CARS2. The flowchart of recruitment is presented in Fig 1.

The total sample consisted of more males than females (p = 0.005), females being less than one-third of males in the autism group (<0.00001). Children with autism were younger than controls (<0.00001), had fewer school years of education (p<0.00001) and belonged to more affluent families (p<0.00001). Age range of the sample was 4–15 years.

### Sensitivity, specificity and positive predictive value (PPV) analysis

Sensitivity and specificity of IASQ was computed for investigating its effectiveness in discriminating autistic and non-autistic children as compared to CARS2. CARS2 has two rating forms (standard–ST, and High Functioning- HF). The children were rated on CARS2-ST, if IQ was 79 and below and on CARS2-HF, if IQ was 80 or above or the child's verbal skills or age was above 6 years (CARS-2 manual, 2012). The data of both the forms was combined to diagnose autism. Specificity and sensitivity of IASQ was calculated compared to CARS-2 (Table 2), for the total sample and separately for the larger NIEPID and the CGOC samples (S1 and S2 Tables).

Since IASQ is a ten-item scale, we calculated these values at every cut-off point. Sensitivity was 97%, specificity 81%, PPV 63%, NPV 99% at cut off 1 while these values were 97%, 92%, 79% and 99% respectively at cut off 2. Positive and negative predictive values and likelihood ratios, at either cut-off (1 or 2) were also acceptable as IASQ is a screening instrument aiming to not miss even one child with autism.

**Likelihood ratio.** In the present study, IASQ cutoffs of 1 and 2 were associated with LR + of 5.98 and 22.42 respectively.

Receiver Operating Characteristic (ROC) curve analysis was conducted to assess the discriminant power of IASQ, using its different cut off levels i.e. 1 to 5 (Fig 2) against diagnostic cut-off of CARS-2. Two curves are presented one with only standard version of CARS2 and other ST and HF versions combined. The overall measure of agreement between criterion CARS2 and IASQ was the area under curve (AUC). This was an estimate of the probability that a randomly selected participant with autism diagnosed with CARS2 would be diagnosed on IASQ at each cut off point than a randomly chosen non-autistic subject. The AUC was used as index of accuracy. The AUC was calculated according to the trapezoidal nonparametric method. The area under the curve for different cut off of IASQ for the total sample is presented in Table 3 and depicted in Fig 2.

IASQ total scores and CARS-2 total scores were compared using Pearson's correlation and plot was drawn. Fig 3 presents the correlation graphically which shows that both scales are positively correlated. The Pearson's correlation between the Z-scores is r = 0.871 (p<0.00001). Fig 3 demonstrates the correlation.

Sensitivity, specificity, Likelihood ratio, NPV and PPV of IASQ were calculated against ISAA also (Table 4). Sensitivity was 97.5%, 97.5% and 95% and specificity was 83.7%, 95.6% and 99.46% at cut-off 1, 2 and 3 respectively.

Comparing diagnosis of autism by ISAA and CARS2: As ISAA had been validated with CARS 1, we examined concordance between the ISAA and CARS-2 in this study. Three combinations with CARS-2 were calculated- CARS2-ST, CARS2-HF and CARS2 ST+HF combined. High concordance rates were present with the combined sample (0.907), CARS2(ST) (0.864) and CARS2 (HF) (-0.954).(Table 5).

ROC Curve was also drawn taking CARS2 diagnosis combined against ISAA diagnosis. The curve is presented in Fig 4. Area under the curve is 0.938.

## Discussion

We compared the Indian Autism Screening Questionnaire (IASQ)–a 10-item screening questionnaire for use of primary care workers in the community- with the Childhood Autism

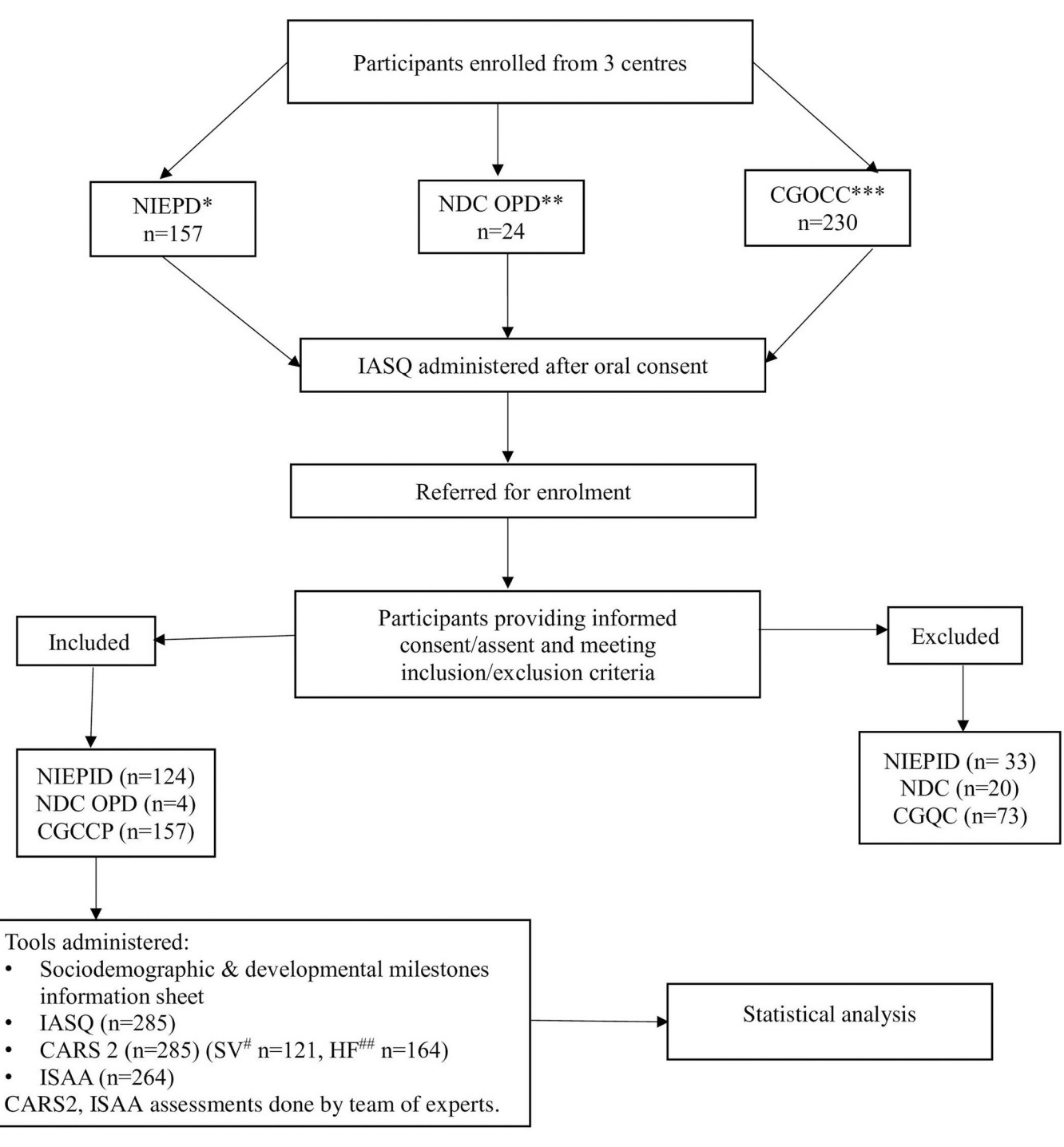

*National Institute for the Empowerment of Persons with Intellectual Disability
** Neuro-Developmental Clinic of Department of Paediatrics
*** Central Government Officers Colony Community
#Standard version
##High Functioning

**Fig 1. Recruitment chart of the participants.**

**Table 2. Sensitivity and specificity of IASQ with CARS2 (ST+HF) (n = 285).**

| Cut off of IASQ | Sensitivity | Specificity | Total Sample (N = 285) | | |
| --- | --- | --- | --- | --- | --- |
| | | | Likelihood Ratio LR+ | NPV | PPV |
| 1 | 0.97 | 0.81 | 5.22 | 0.99 | 0.63 |
| 2 | 0.97 | 0.92 | 11.60 | 0.99 | 0.79 |
| 3 | 0.96 | 0.95 | 20.58 | 0.99 | 0.87 |
| 4 | 0.94 | 0.97 | 28.96 | 0.98 | 0.90 |
| 5 | 0.92 | 0.98 | 39.37 | 0.98 | 0.93 |
| 6 | 0.86 | 0.99 | 61.98 | 0.95 | 0.96 |
| 7 | 0.80 | 0.99 | 57.33 | 0.94 | 0.98 |
| 8 | 0.60 | 1.00 | Infinity | 0.88 | 1.00 |
| 9 | 0.36 | 1.00 | Infinity | 0.83 | 1.00 |
| 10 | 0.04 | 1.00 | Infinity | 0.76 | 1.00 |

Rating Scale version 2 (CARS2) in a special outpatient's clinic, a rehabilitation institute and the community. There were more boys than girls in the Autism group. Autism affects females less frequently than males [30]. Significantly more autistic children were studying in special schools.

The present study was carried out to validate the IASQ against the CARS2. The IASQ was developed from ISAA and validation was carried out initially with the same instrument. Since the IASQ was derived from the ISAA (which had itself been compared to the first version of the CARS), we compared both IASQ and ISAA with CARS2 in different samples. CARS2, an updated version of CARS is a widely used rating scale for Autism Spectrum Disorder.

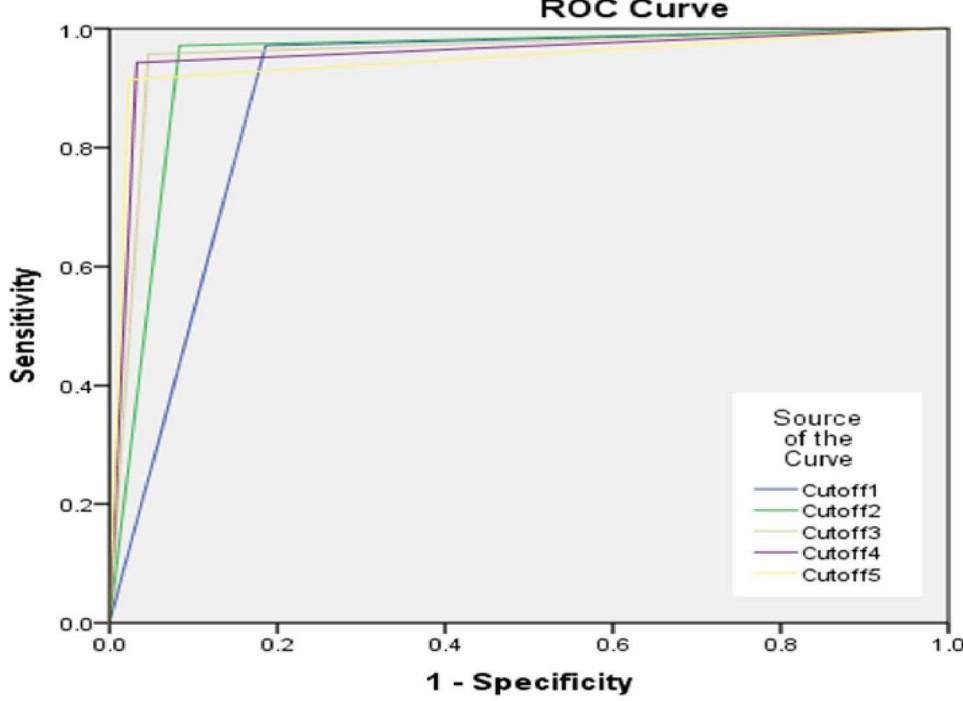

Diagonal segments are produced by ties.

**Fig 2. Receiver operating characteristic (ROC) curve analysis of IASQ against CARS2 (n = 285).**

**Table 3. Results of ROC curve analysis.**

| IASQ and CARS2 ST +CARS2 HF combined (n = 285) | | | | | |
|---|---|---|---|---|---|
| IASQ Cut off | Area | Standard Error | Significance level | Asymptotic 95% Confidence Interval | |
| | | | | Lower Bound | Upper Bound |
| 1 | 0.893 | 0.020 | 0.000 | 0.853 | 0.932 |
| 2 | 0.944 | 0.016 | 0.000 | 0.912 | 0.975 |
| 3 | 0.955 | 0.016 | 0.000 | 0.923 | 0.987 |
| 4 | 0.955 | 0.018 | 0.000 | 0.921 | 0.990 |
| 5 | 0.946 | 0.020 | 0.000 | 0.906 | 0.985 |
| 6 | 0.922 | 0.025 | 0.000 | 0.872 | 0.971 |
| 7 | 0.893 | 0.029 | 0.000 | 0.836 | 0.950 |
| 8 | 0.798 | 0.037 | 0.000 | 0.724 | 0.871 |
| 9 | 0.679 | 0.042 | 0.000 | 0.597 | 0.761 |
| 10 | 0.521 | 0.041 | 0.590 | 0.442 | 0.601 |

Sensitivity of IASQ was 97% at both cut-off 1 and 2. It increased to 99% at cut-off 3 against CARS2. However, specificity was 92% at cut-off 2. Thus if we need both sensitivity and specificity cut-off 2 is appropriate. The aim of IASQ was screening and cut-off 1–3 are acceptable for use in community to identify possible cases depending on the requirement.

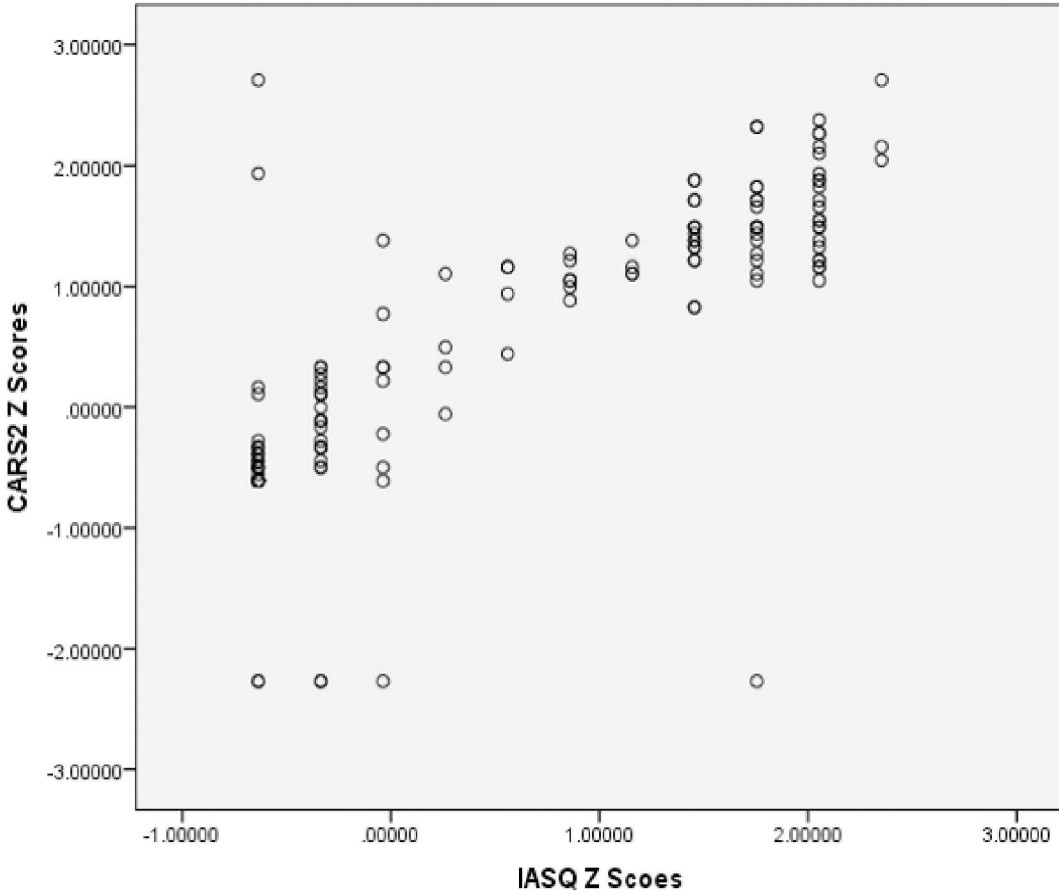

**Fig 3. Correlation between IASQ and CARS2 (standard version + high functioning) (n = 285).**

**Table 4. Sensitivity and specificity of IASQ against ISAA.** (n = 264).

| Cut off of IASQ | Sensitivity | Specificity | Likelihood Ratio LR+ | NPV* | PPV** |
|---|---|---|---|---|---|
| 1 | 97.50 | 83.70 | 5.98 | 0.01 | 0.14 |
| 2 | 97.50 | 95.65 | 22.42 | 0.01 | 0.04 |
| 3 | 95.00 | 99.46 | 174.8 | 0.01 | 0.02 |
| 4 | 91.25 | 100.00 | Infinity | 0.04 | 0.00 |
| 5 | 86.25 | 100.00 | Infinity | 0.06 | 0.00 |
| 6 | 78.75 | 100.00 | Infinity | 0.08 | 0.00 |
| 7 | 73.75 | 100.00 | Infinity | 0.10 | 0.00 |
| 8 | 53.75 | 100.00 | Infinity | 0.17 | 0.00 |
| 9 | 31.25 | 100.00 | Infinity | 0.23 | 0.00 |
| 10 | 3.75 | 100.00 | Infinity | 0.30 | 0.00 |

*Negative Predictive values

**Positive Predictive Value

CARS2 has two different scales, Standard (ST) and High Functioning (HF). The areas under the curve against CARS2(ST), CARS2 (ST+HF) and CARS2(HF) were calculated separately. CARS2 identifies high functioning cases separately and CARS2 (ST) is not administered if IQ is higher than 79. IASQ being a screening tool does not discriminate between high functioning and lower functioning children. Area under the curve against ST version was 0.872. When ST and HF were combined the area increased to 0.893 at cut off 1. The cut-off 2 had greater AUC which suggests that IASQ is quite a sensitive instrument and can screen in children with autism. The sensitivity is in accordance with our earlier study of psychometric properties of IASQ against ISAA which was 99% at cut-off 1 with specificity of 62%. The specificity against CARS2 is higher i.e. 0.81. The Trivandrum Autism Behaviour Checklist (TABC) [31] is the other Indian scale validated and showing agreement with CARS [32]. IASQ Z-scores had high correlation with CARS2 Z-scores suggesting that both measure similar attributes. We separately calculated sensitivity and specificity for the smaller sample from NIEPID which is also quite high suggesting the significant psychometric properties of the screening instrument.

We also calculated sensitivity and specificity of IASQ against ISAA and it is quite high for cut off 1, 2 and 3. Our earlier manuscript reported that at a cut-off of 1, sensitivity was 99%, specificity 62%, Positive Predictive Value 81%, and Negative Predictive Value 95%. In the present, more diverse sample, sensitivity and specificity are 97.5% and 83.7% respectively [11]. Although the current samples were different from our earlier clinic sample, results remained fairly consistent and acceptable. ISAA and CARS2 were compared and high concordance was found between these two. Both the scales identify similar symptoms in autism.

Considering the sensitivity and specificity of the IASQ, it emerges as a good screening tool. Cut off 1 and 2 consistently showed high psychometric values and can be used with confidence in the community. IASQ is able to identify high functioning autistic children as well. Mukherjee et al. [33, 34] compared the ISAA with CARS2 in a sample below three years of age (the

**Table 5. Concordance of ISAA and CARS2; and Area under ROC (n = 264).**

| Concordance variable | Kappa value (p-value) | Level of Significance |
|---|---|---|
| CARS2 (ST+HF) vs ISAA (n = 264) | 0.907 | <0.00001 |
| CARS2 (ST) (n = 1118)* | 0.864 | <0.00001 |
| CARS2 (HF) (n = 146) | 0.954 | <0.00001 |

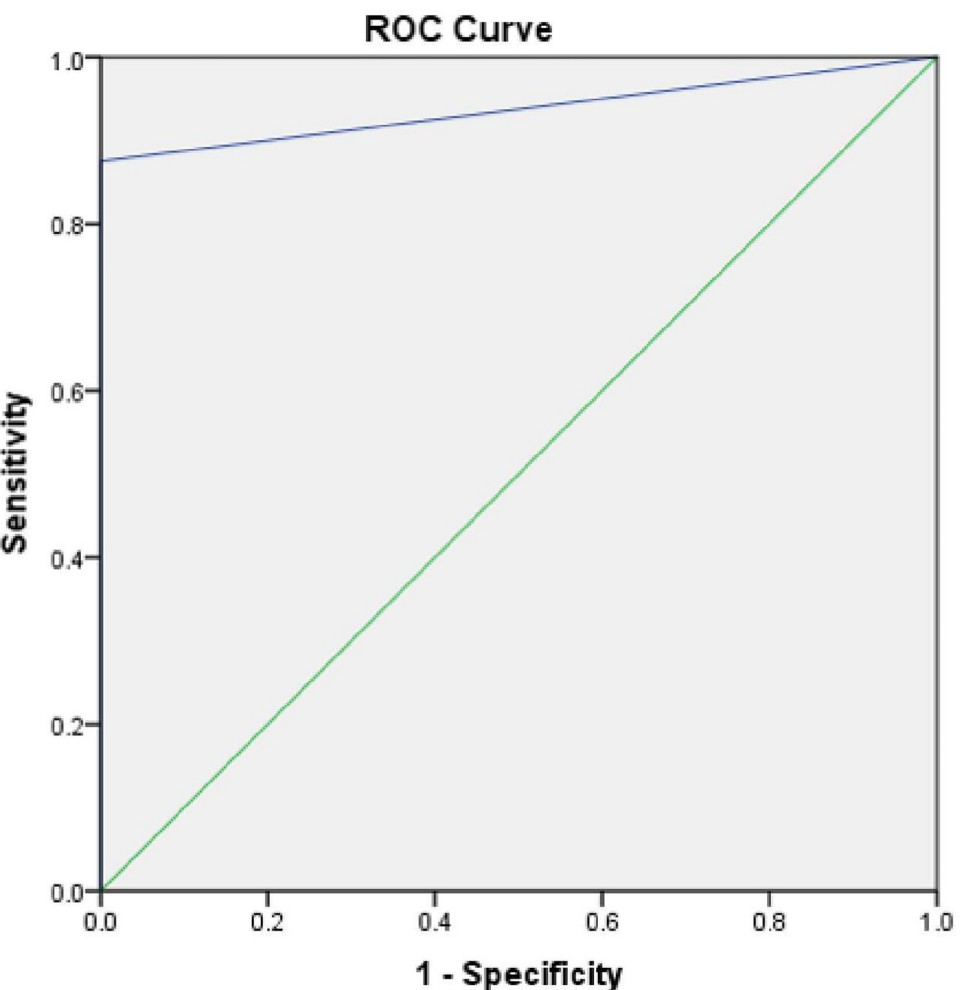

**Fig 4. Receiver operating characteristic (ROC) curve analysis of ISAA against CARS2 (n = 264).**

ISAA was developed for children above 3 years of age). However, in children above 3 years of age, CARS2 and IASQ- although developed for different purposes-are comparable. The ISAA itself has been developed and tested for children above 3 years of age, while CARS2 is suitable for children above 2 years of age [9, 22, 35]. As for diagnostic likelihood ratio in the present study, IASQ cutoffs of 1 and 2 were associated with LR+s of 5.98 and 22.42 respectively which are high and acceptable. While its non-linear nature is a known drawback of the diagnostic likelihood ratio or DLR (denoted by LR+ in this paper), it provides the best measure of the discriminative power of a test by summarizing both its sensitivity and specificity. Furthermore, unlike sensitivity or specificity, DLR remains unaffected by the prevalence of the disease. The LR+ of the IASQ are also acceptable.

## Conclusion

IASQ has high sensitivity and specificity against CARS2. It can be used for screening of children with autism in community at a cut off of 1 or 2. IASQ can identify high functioning children with autism. ISAA and CARS 2 are also concordant with each other.

## Supporting information

**S1 Table. Sensitivity and specificity of IASQ with ISAA (n = 124).**
(DOCX)

**S2 Table. Sensitivity and specificity of IASQ with ISAA (n = 157).**
(DOCX)

## Acknowledgments

We thank Dr. Soumya Swaminathan (Secretary, Dept. of Health Research, DHR at the time of project development), Dr. Balram Bhargav (Secretary DHR), and Dr. Harpreet Singh, ICMR (for developing i Mann database). We thank the faculty of 'Cross-Fertilized Research Training for New Investigators in India and Egypt'). We thank the National Coordinating Unit for logistic support and the ICMR Data Management Unit for designing the database.

## Author Contributions

**Conceptualization:** Satabdi Chakraborty, Triptish Bhatia, Jaspreet S. Brar, Smita Neelkanth Deshpande.

**Data curation:** Nitin Antony, Aratrika Roy, Vandana Shriharsh.

**Formal analysis:** Triptish Bhatia, Satish Iyengar, Vishwajit L. Nimgaonkar.

**Investigation:** Nitin Antony, Aratrika Roy, Vandana Shriharsh, Amrita Sahay.

**Methodology:** Satabdi Chakraborty, Triptish Bhatia, Smita Neelkanth Deshpande.

**Project administration:** Satabdi Chakraborty, Smita Neelkanth Deshpande.

**Resources:** Satabdi Chakraborty, Amrita Sahay.

**Software:** Jaspreet S. Brar.

**Supervision:** Satabdi Chakraborty, Triptish Bhatia, Smita Neelkanth Deshpande.

**Validation:** Satish Iyengar, Vishwajit L. Nimgaonkar.

**Visualization:** Ravinder Singh.

**Writing – original draft:** Triptish Bhatia, Nitin Antony.

**Writing – review & editing:** Satabdi Chakraborty, Jaspreet S. Brar, Satish Iyengar, Ravinder Singh, Vishwajit L. Nimgaonkar, Smita Neelkanth Deshpande.

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
