## [Decision Letter · Decision Letter 0]

12 May 2022

PONE-D-21-32345Comparing the Indian Autism Screening Questionnaire (IASQ) and the Indian Scale for Assessment of Autism (ISAA) with the Childhood Autism Rating Scale–Second Edition (CARS2) in Indian settingPLOS ONE

Dear Dr. Deshpande,

Thank you for submitting your manuscript to PLOS ONE. After careful consideration, we feel that it has merit but does not fully meet PLOS ONE’s publication criteria as it currently stands. Therefore, we invite you to submit a revised version of the manuscript that addresses the points raised during the review process.

Your manuscript has been assessed by an expert reviewer, whose comments are appended below. The reviewer requests clarification on one aspect of your analysis, and suggests you have your manuscript copyedited to correct typographical errors. Please respond to all of the points carefully in your response to reviewers, and revise your manuscript accordingly.  Please note that we have only been able to secure a single reviewer to assess your manuscript. We are issuing a decision on your manuscript at this point to prevent further delays in the evaluation of your manuscript. Please be aware that the editor who handles your revised manuscript might find it necessary to invite additional reviewers to assess this work once the revised manuscript is submitted. However, we will aim to proceed on the basis of this single review if possible.

We look forward to receiving your revised manuscript.

Kind regards,

Joseph Donlan

Editorial Office

PLOS ONE

**Journal requirements:**

“We thank Dr. Soumya Swaminathan (then Secretary, Dept. of Health Research, DHR), Dr. Balram Bhargav (Secretary DHR), and Dr. Harpreet Singh, ICMR. We thank the faculty of ‘Cross-Fertilized Research Training for New Investigators in India and Egypt’ (D43 TW009114, HMSC File No. Indo-Foreign/35/M/2012-NCD-1, funded by Fogarty International Centre, NIH).   We thank the National Coordinating Unit for logistic support and the ICMR Data Management Unit for designing the database.  The content of this manuscript is solely the responsibility of the authors and does not necessarily represent the official views of NIH or ICMR. NIH and ICMR had no role in the design and conduct of the study; collection, management, analysis, and interpretation of the data; preparation, review, or approval of the manuscript; and decision to submit the manuscript for publication.”

5. Please include a caption for figure 1.

**Reviewers' comments:**

Reviewer's Responses to Questions

**Comments to the Author**

1. Is the manuscript technically sound, and do the data support the conclusions?

Reviewer #1: Yes

2. Has the statistical analysis been performed appropriately and rigorously? 

Reviewer #1: Yes

3. Have the authors made all data underlying the findings in their manuscript fully available?

Reviewer #1: Yes

4. Is the manuscript presented in an intelligible fashion and written in standard English?

Reviewer #1: Yes

5. Review Comments to the Author

Reviewer #1: This paper dealing with the psychometric properties of two autism screening scales validated against the CARS 2 has several strengths and one or two weaknesses. The latter concerns the english language which could be improved and that there is one omission when it comes to the analytic plan.

The research procedureres seem rigorous based on the descriptions and the data analyses are performed in an expert way. The concordant validity and discriminant validity are well described and shows adequate properties.

There data when it comes to psychometrics: i.e. sensitivity, specificity, PPV and NPV as well as the ROC-analysis are to the point. The omission is, I think that the likelihood ratio (diagnostic likelihood ratio (DLR) could be used in a clearer way and not just represented in a table. This is important when the diagnostic task involves a low prevalence disorder such a as autism. The reason I'd like to highlight it is that using Bayesian thinking even reasonably high DLR that are shown based on a ISAAQ cut-off of 2 or 3 points (roughly 11 and 20 OR) render quite a weak signal as what to do with a positive cut-off. When the scales are used in a screening situation where the base rate of autism is close to 1%, the odds for an autism diagnosis does not raise higher than 20 respectively 30% from a score above the cut-off. These odds are roughly calulated based on a nomogram as described by Youngstrom (e.g., 1. Youngstrom E. Future Directions in Psychological Assessment: Combining Evidence-Based Medicine Innovations with Psychology's Historical Strengths to Enhance Utility. Journal of Clinical Child and Adolescent Psychology. 2013;42(1):139-59.

Thus, I would like authors to argue in a more explicit way on what scores should be used and and what needs to be done following a positive screening outcome.

As a summary, however, I like the paper and I feel its utility would be enhanced if my suggestions for improvements are used.

6. PLOS authors have the option to publish the peer review history of their article (what does this mean?). If published, this will include your full peer review and any attached files.

Reviewer #1: No

---

## [Author Response · Author response to Decision Letter 0]

25 May 2022

Reply to reviewers’ and editorial comments:

Editorial comments

Comment 1. Please ensure that your manuscript meets PLOS ONE's style requirements, including those for file naming. The PLOS ONE style templates can be found at

Reply1: Thank you. We have checked these and made the necessary changes.

Comment 2. We note that the grant information you provided in the ‘Funding Information’ and ‘Financial Disclosure’ sections do not match. When you resubmit, please ensure that you provide the correct grant numbers for the awards you received for your study in the ‘Funding Information’ section.

Reply2: Thank you for the guidance. We have changed the funding information as under:

Funding: This research was funded by the Indian Council of Medical Research Task Force for Capacity Building for National Mental Health Programme (File No No. 5/4-4/151/M/2017/NCD-I) and the Fogarty International Centre, NIH (‘Cross-Fertilized Research Training for New Investigators in India and Egypt’, D43 TW009114 Fogarty International Centre, NIH and HMSC File No. Indo-Foreign/35/M/2012-NCD-1).The content of this manuscript is solely the responsibility of the authors and does not necessarily represent the official views of Fogarty International Centre, NIH (Cross-Fertilized Research Training for New Investigators in India and Egypt’ (D43 TW009114) or ICMR had no role in the design and conduct of the study; collection, management, analysis, and interpretation of the data; preparation, review, or approval of the manuscript; and decision to submit the manuscript for publication

Comment 3. Thank you for stating the following in the Acknowledgments Section of your manuscript:

“We thank Dr. Soumya Swaminathan (then Secretary, Dept. of Health Research, DHR), Dr. Balram Bhargav (Secretary DHR), and Dr. Harpreet Singh, ICMR. We thank the faculty of ‘Cross-Fertilized Research Training for New Investigators in India and Egypt’ (D43 TW009114, HMSC File No. Indo-Foreign/35/M/2012-NCD-1, funded by Fogarty International Centre, NIH). We thank the National Coordinating Unit for logistic support and the ICMR Data Management Unit for designing the database. The content of this manuscript is solely the responsibility of the authors and does not necessarily represent the official views of NIH or ICMR. NIH and ICMR had no role in the design and conduct of the study; collection, management, analysis, and interpretation of the data; preparation, review, or approval of the manuscript; and decision to submit the manuscript for publication.”

Reply3: Thank you for clarifying this. We have deleted the funding information in the manuscript. 

The funding statement needs to be changed as follows:

This research was funded by the Indian Council of Medical Research Task Force for Capacity Building for National Mental Health Programme (File No No. 5/4-4/151/M/2017/NCD-I) and the Fogarty International Centre, NIH (‘Cross-Fertilized Research Training for New Investigators in India and Egypt’, D43 TW009114 Fogarty International Centre, NIH and HMSC File No. Indo-Foreign/35/M/2012-NCD-1The content of this manuscript is solely the responsibility of the authors and does not necessarily represent the official views of Fogarty International Centre, NIH (Cross-Fertilized Research Training for New Investigators in India and Egypt’ (D43 TW009114) or ICMR had no role in the design and conduct of the study; collection, management, analysis, and interpretation of the data; preparation, review, or approval of the manuscript; and decision to submit the manuscript for publication.

Comment 4. Please amend your list of authors on the manuscript to ensure that each author is linked to an affiliation. Authors’ affiliations should reflect the institution where the work was done (if authors moved subsequently, you can also list the new affiliation stating “current affiliation:….” as necessary).

Reply 4: All authors are linked to affiliation

Comment 5. Please include a caption for figure 1.

Reply 5: I have included the caption of all figures in the manuscript.

Comment 6: Please review your reference list to ensure that it is complete and correct. If you have cited papers that have been retracted, please include the rationale for doing so in the manuscript text, or remove these references and replace them with relevant current references. Any changes to the reference list should be mentioned in the rebuttal letter that accompanies your revised manuscript. If you need to cite a retracted article, indicate the article’s retracted status in the References list and also include a citation and full reference for the retraction notice.

[Note: HTML mark-up is below. Please do not edit.]

Reply 6: Reference list is checked. It is complete and is according to PLOS ONE's style.

Reviewers' comments:

Reviewer's Responses to Questions

Comments to the Author

1. Is the manuscript technically sound, and do the data support the conclusions?

Reviewer #1: Yes

2. Has the statistical analysis been performed appropriately and rigorously?

Reviewer #1: Yes

3. Have the authors made all data underlying the findings in their manuscript fully available?

Reviewer #1: Yes

4. Is the manuscript presented in an intelligible fashion and written in standard English?

Reviewer #1: Yes

Reply: Thank you very much for this endorsement, respected reviewer. 

5. Review Comments to the Author

Reviewer #1: 

Query 1: This paper dealing with the psychometric properties of two autism screening scales validated against the CARS 2 has several strengths and one or two weaknesses. The latter concerns the English language which could be improved and that there is one omission when it comes to the analytic plan.

The research procedures seem rigorous based on the descriptions and the data analyses are performed in an expert way. The concordant validity and discriminant validity are well described and shows adequate properties.

Reply 1: Thank you very much for the appreciation and encouraging remarks. We have rechecked the manuscript for English language and simplified text to the extent possible. 

Query 2.There data when it comes to psychometrics: i.e. sensitivity, specificity, PPV and NPV as well as the ROC-analysis are to the point. The omission is, I think that the likelihood ratio (diagnostic likelihood ratio (DLR) could be used in a clearer way and not just represented in a table. This is important when the diagnostic task involves a low prevalence disorder such a as autism. The reason I'd like to highlight it is that using Bayesian thinking even reasonably high DLR that are shown based on a ISAAQ cut-off of 2 or 3 points (roughly 11 and 20 OR) render quite a weak signal as what to do with a positive cut-off. When the scales are used in a screening situation where the base rate of autism is close to 1%, the odds for an autism diagnosis does not raise higher than 20 respectively 30% from a score above the cut-off. These odds are roughly calulated based on a nomogram as described by Youngstrom (e.g., 1. Youngstrom E. Future Directions in Psychological Assessment: Combining Evidence-Based Medicine Innovations with Psychology's Historical Strengths to Enhance Utility. Journal of Clinical Child and Adolescent Psychology. 2013;42(1):139-59.

Thus, I would like authors to argue in a more explicit way on what scores should be used and what needs to be done following a positive screening outcome.

As a summary, however, I like the paper and I feel its utility would be enhanced if my suggestions for improvements are used.

Reply 2:. We thank the reviewer for liking the manuscript. The reviewer correctly points out the non-linear nature of the diagnostic likelihood ratio or DLR (denoted by LR+ in the manuscript). While this is a known drawback of this metric, it provides the best measure of the discriminative power of the test by summarizing both its sensitivity and specificity. Furthermore, unlike sensitivity or specificity, DLR remains unaffected by the prevalence of the disease. We have therefore added clarification in the manuscript to aid the reader in interpreting the DLR and its utility in clinical decision making.

We have added text to explain this matter in various sections as follows:

Methods:

Test psychometrics:

…. Likelihood ratios provide the best guide to clinicians to determine whether a diagnostic test usefully changes the odds that a disease exists. By convention, a positive diagnostic likelihood ratio (LR+) between 1 to 5 is associated with a small increase in post-test odds whereas values between 5 and 10 with a moderate increase and values above 10 with a large increase in post-test odds. Both ranges, 5 to 10 and >10 are considered acceptable for clinical decision making. To further examine the effects that these LR+s may have on the likelihood of disease a Fagan’s Nomogram may be used after first converting the pre-test odds to a pre-test probability and then drawing a straight line from the pre-test probability through the LR+ to the corresponding post-test probability [25, 26]. Alternatively, interactive online calculators, for example the diagnostic test calculator available at the University of Illinois, Chicago, USA (http://araw.mede.uic.edu/cgi-bin/testcalc.pl) may be used to calculate LR+ and post-test probabilities [27]. The latter two steps were not followed in this study.

Results:

In the present study, IASQ cut-offs of 1 and 2 were associated with LR+s of 5.98 and 22.42 respectively.

Discussion:

…………. As for diagnostic likelihood ratio in the present study, IASQ cutoffs of 1 and 2 were associated with LR+s of 5.98 and 22.42 respectively which are high and acceptable. While its non-linear nature is a known drawback of the diagnostic likelihood ratio or DLR (denoted by LR+ in this paper), it provides the best measure of the discriminative power of a test by summarizing both its sensitivity and specificity. Furthermore, unlike sensitivity or specificity, DLR remains unaffected by the prevalence of the disease. The LR+ of the IASQ are also acceptable.

6. PLOS authors have the option to publish the peer review history of their article (what does this mean?). If published, this will include your full peer review and any attached files.

Do you want your identity to be public for this peer review? For information about this choice, including consent withdrawal, please see our Privacy Policy.

Reviewer #1: No

---

## [Decision Letter · Decision Letter 1]

1 Aug 2022

PONE-D-21-32345R1Comparing the Indian Autism Screening Questionnaire (IASQ) and the Indian Scale for Assessment of Autism (ISAA) with the Childhood Autism Rating Scale–Second Edition (CARS2) in Indian settingPLOS ONE

Dear Dr. Deshpande,

Thank you for submitting your manuscript to PLOS ONE. After careful consideration, we feel that it has merit but does not fully meet PLOS ONE’s publication criteria as it currently stands. Therefore, we invite you to submit a revised version of the manuscript that addresses the points raised during the review process.

We look forward to receiving your revised manuscript.

Kind regards,

Tord Ivarsson

Guest Editor

PLOS ONE

Journal Requirements:

Additional Editor Comments (if provided):

Thank your for your re-submission based on my review. I have now accepted the role as guest editor for your manuscript. I have in that role sent the revised manuscript to a second reviewer who is very knowledgable in psychometrics and in neuropsychiatric disorders. He has recommended a few, easy to implement additions to your description of the sample. As soon as these changes have been done I will evaluate the manuscript as quickly as I can.

However, one of these suggestions, that you should add information avout the severity of the autistic disorder may not be available to you. In that case, you may just add a comment about the lack of data in this respect, and I will not hold this against you in my final decisions.

Both the second reviewer and I think that you have done an excellent job with regard to the analytic plan and the statistical analyses, and see no need for further work within that area.

Kind regards,

Tord Ivarsson

Special guest editor.

Reviewers' comments:

Reviewer's Responses to Questions

**Comments to the Author**

1. If the authors have adequately addressed your comments raised in a previous round of review and you feel that this manuscript is now acceptable for publication, you may indicate that here to bypass the “Comments to the Author” section, enter your conflict of interest statement in the “Confidential to Editor” section, and submit your "Accept" recommendation.

Reviewer #2: (No Response)

2. Is the manuscript technically sound, and do the data support the conclusions?

Reviewer #2: Yes

3. Has the statistical analysis been performed appropriately and rigorously? 

Reviewer #2: Yes

4. Have the authors made all data underlying the findings in their manuscript fully available?

Reviewer #2: Yes

5. Is the manuscript presented in an intelligible fashion and written in standard English?

Reviewer #2: Yes

6. Review Comments to the Author

Reviewer #2: Thank you for inviting me to review this article. I find the study well designed and analyses excellent. Background and rationale well described, and discussion of results clear and useful. I agree with the comments of previous reviewers.

I have only a few further comments/suggestions:

In the "Sample" section for clarity it could be mentioned that the sample consists of 70 children with autism, and 215 children without autism, and ref to Table 1.

How were the autism diagnoses ascertained? By team assessments? Child psychiatrists /pediatricians /psychologists? According to DSM-5 criteria I presume?

This information would support the quality of the sample.

And perhaps if there are data on the severity of the autism (Level 1, 2 or 3?) that could be added.

In the section "Inclusion and exclusion criteria" it is stated that children with known comorbid or psychiatric disorders were excluded. Which disorders?

7. PLOS authors have the option to publish the peer review history of their article (what does this mean?). If published, this will include your full peer review and any attached files.

Reviewer #2: No

---

## [Author Response · Author response to Decision Letter 1]

5 Aug 2022

Dr. Smita N. Deshpande

Professor

Dept. of Psychiatry, De-addiction Services,

Centre of Excellence in Mental Health,

Atal Bihari Vajpayee Institute of Medical Sciences & 

Dr Ram Manohar Lohia Hospital,

Banga Bandhu Sheikh Mujib Road,

New Delhi 110001

Phone: 011 23404269, 23404363. 

Email: smitadeshp@gmail.com

5th August, 2022

To,

The Editor-in-Chief

Plos One.

Ref: Resubmission of our manuscript after incorporating peer review suggesitons entitled “Comparing the Indian Autism Screening Questionnaire (IASQ) and the Indian Scale for Assessment of Autism (ISAA) with the Childhood Autism Rating Scale–Second Edition (CARS2) in Indian settings” [PONE-D-21-32345R1].

Dear Dr.Tord Ivarsson,

Thank you for reviewing our manuscript and getting it reviewed by an expert. The respected peer reviewer appreciated our work and we have addressed his two small comments. We thank the reviewers for very encouraging and appreciative remarks. 

We hope that the manuscript is appropriate for publishing. I request that a positive decision is reached soon and that this paper will be published at the earliest.

Thank you.

Yours sincerely,

Smita N Deshpande, MD

Response to the comments

Comments to the Author

1. If the authors have adequately addressed your comments raised in a previous round of review and you feel that this manuscript is now acceptable for publication, you may indicate that here to bypass the “Comments to the Author” section, enter your conflict of interest statement in the “Confidential to Editor” section, and submit your "Accept" recommendation.

Reviewer #2: (No Response)

Thank you for accepting our replies to your comments.

2. Is the manuscript technically sound, and do the data support the conclusions?

Reviewer #2: Yes

Reply: Appreciate your comment.

3. Has the statistical analysis been performed appropriately and rigorously?

Reviewer #2: Yes

Reply: Thanks for positive comment.

4. Have the authors made all data underlying the findings in their manuscript fully available?

Reviewer #2: Yes

5. Is the manuscript presented in an intelligible fashion and written in standard English?

Reviewer #2: Yes

6. Review Comments to the Author

Reviewer #2: Thank you for inviting me to review this article. I find the study well designed and analyses excellent. Background and rationale well described, and discussion of results clear and useful. I agree with the comments of previous reviewers.

Thank you very much for your encouraging comments and appreciation.

I have only a few further comments/suggestions:

Query 1: In the "Sample" section for clarity it could be mentioned that the sample consists of 70 children with autism, and 215 children without autism, and ref to Table 1.How were the autism diagnoses ascertained? By team assessments? Child psychiatrists /pediatricians /psychologists? According to DSM-5 criteria I presume?

This information would support the quality of the sample.

And perhaps if there are data on the severity of the autism (Level 1, 2 or 3?) that could be added.

Reply1: We have added the following text: in sample section.

“The sample consisted of 70 children with autism, and 215 children without autism (Table1). The diagnosis was ascertained by CARS 2 ST and HF versions. CARS 2 diagnosis fulfills DSM V criteria.”

Query 2:In the section "Inclusion and exclusion criteria" it is stated that children with known comorbid or psychiatric disorders were excluded. Which disorders?

Reply 2: Thank you for this important observation. We added the following text in inclusion and exclusion criteria

Children with other comorbid psychiatric disorders, for example psychotic disorders, ADHD and intellectual disabilities were excluded. An asessment of illness severity was not available on most participants and therefore not analyzed.

7. PLOS authors have the option to publish the peer review history of their article (what does this mean?). If published, this will include your full peer review and any attached files.

Do you want your identity to be public for this peer review? For information about this choice, including consent withdrawal, please see our Privacy Policy.

Reviewer #2: No

---

## [Editor Report · Decision Letter 2]

16 Aug 2022

Comparing the Indian Autism Screening Questionnaire (IASQ) and the Indian Scale for Assessment of Autism (ISAA) with the Childhood Autism Rating Scale–Second Edition (CARS2) in Indian setting

PONE-D-21-32345R2

Dear Dr. Deshpande,

We’re pleased to inform you that your manuscript has been judged scientifically suitable for publication and will be formally accepted for publication once it meets all outstanding technical requirements.

Kind regards,

Tord Ivarsson

Guest Editor

PLOS ONE
---

## [Editor Report · Acceptance letter]

31 Aug 2022

PONE-D-21-32345R2 

Comparing the Indian Autism Screening Questionnaire (IASQ) and the Indian Scale for Assessment of Autism (ISAA) with the Childhood Autism Rating Scale–Second Edition (CARS2) in Indian settings 

Dear Dr. Deshpande:

I'm pleased to inform you that your manuscript has been deemed suitable for publication in PLOS ONE. Congratulations! Your manuscript is now with our production department. 

Kind regards, 

on behalf of

Dr. Tord Ivarsson 

Guest Editor

PLOS ONE